# A Novel Strategy to Model Age-Related Cancer for Elucidation of the Role of Th17 Inflammaging in Cancer Progression

**DOI:** 10.3390/cancers14215185

**Published:** 2022-10-22

**Authors:** Qiuyang Zhang, S. Michal Jazwinski

**Affiliations:** 1Department of Structural & Cellular Biology, Tulane University School of Medicine, New Orleans, LA 70112, USA; 2Tulane Center for Aging, Tulane University School of Medicine, New Orleans, LA 70112, USA; 3Tulane Cancer Center, Tulane University School of Medicine, New Orleans, LA 70112, USA; 4Deming Department of Medicine, Tulane University School of Medicine, New Orleans, LA 70112, USA

**Keywords:** Th17, inflammaging, aging, mouse model

## Abstract

**Simple Summary:**

Cancer is a disease of aging, but how aging contributes to cancer remains unclear. A mouse model of cancer that allows the spatially and temporally controlled initiation of tumors is critical to understanding the molecular mechanisms of the complex processes and related gene pathways of biological aging that play a role in cancer initiation and progression. One of the hallmarks of aging is chronic inflammation, often called inflammaging. We have implicated aging-related inflammation in prostate cancer, a major aging malignancy. We present our novel strategy for the generation of age-related cancer models in mice and discuss how we use these prostate cancer mouse models, generated at different ages, to elucidate mechanisms contributing to development of tumors during aging.

**Abstract:**

Cancer is a disease of aging, but most studies on cancer are in young but not aged animal models, and cancer clinical trials are rarely performed in older adults. Recognition of the connections between aging and cancer and improvement of treatment for elderly cancer patients has become one of the most critical medical issues with the global increase in the elderly population. Mouse models are essential experimental tools for understanding the molecular mechanisms of complex processes and related gene pathways of biological aging. However, few mouse models can be used to understand the role of aging in cancer development and the underlying mechanisms. One of the hallmarks of aging is chronic inflammation, often called inflammaging. This is our rationale for examining the role of aging-related inflammation in prostate cancer, a major aging malignancy. We have now developed a novel method to generate age-related cancer models in mice to better understand how age impacts cancer initiation and progression in the natural aging process. We discuss its application to elucidate some of the contributing mechanisms.

## 1. Introduction

With the global increase in the elderly population, improving the treatment for various aging-related diseases, including cancers, has become a critical current medical issue [1]. According to the World Health Organization, in most developed countries, life expectancy exceeds 80 years. The process of becoming older is a complex scenario influenced by the interaction of various genetic and environmental factors. Aging represents the single most significant risk factor for cancer development, and the US National Cancer Institute’s Surveillance Epidemiology and End Results (SEER) database shows that 40.2% of men and 38.5% of women will develop cancer throughout their lives [2]. Among these, 17% of men and 15% of women die from cancer. More than half of cancers occur in individuals older than 70, of which 32.9% are men and 26.5% women [2]. Cancer is often defined as a disease of aging, but it is rarely studied in aged animal models, and older adults are often insufficiently represented in cancer clinical trials. Despite the clear epidemiology of this disease and similarities between aging and cancer biology, most preclinical studies on cancer ignore the aging dimension of the disease. Typically, mouse studies use young animals of 6–8 weeks of age [3], the equivalent of approximately 15–20-year-old humans. However, most cancers start around the age of 50 and increase in prevalence as we become older [2].

A longer tumor latency suggests that involvement of the target molecule is age dependent in prostate tumorigenesis. Unfortunately, few studies have defined how aging impacts tumor initiation or progression. Multiple issues might explain this lack of research: difficulties separating the aging effect from tumorigenesis, the commitment of time to complete a project, and the cost and the effort required to model aging. The first of these issues also is a confounder that must be accounted for. Furthermore, in most genetically engineered mouse models (GEMM), tumorigenesis is too rapid to permit aging-related comparisons. In addition, we must consider the complex parameters of the entire in vivo biological system and not just the tumor alone. The lack of good GEMM for studying aging and prostate tumorigenesis has hindered progress in this area. We introduce our study here to provide an example of analyzing age-related tumorigenesis in the prostate gland.

## 2. The Importance of Examining the Effect of Aging in Cancer

Aging represents a series of biological processes, including direct damage, accumulation of changes, cellular waste, errors, imperfect repairs, and responses [4,5]. These processes result in the familiar signs of aging and ultimately lead to the development of age-related diseases, such as Alzheimer’s disease, diabetes, cardiovascular disease, stroke, and cancer, eventually killing us. There are multiple perspectives on aging; the definition of nine cellular and molecular hallmarks of aging proposed by López-Otín and colleagues in 2013 [5] has been instrumental in guiding and advancing research on the biology of aging. These nine hallmarks include genomic instability, telomere attrition, epigenetic alterations, mitochondrial dysfunction, loss of proteostasis, deregulated nutrient-sensing, cellular senescence, stem cell exhaustion, and altered intercellular communication. However, these hallmarks have recently been criticized for being insufficient in serving as a causative paradigm of aging [6], though they have been shown to map to age-related diseases [7]. Therefore, a recent research symposium, “New Hallmarks of Ageing,” was held, focusing on novel findings and the recontextualization of the nine hallmarks of aging. The new proposed hallmarks of aging [4] include autophagy, microbiome disturbance, altered mechanical properties, splicing dysregulation, and inflammation, among other emerging ones. Therefore, the amalgamation of the original and “new” hallmarks of aging will provide a more comprehensive explanation of aging and age-related diseases. Importantly, inflammation, or age-dependent chronic inflammation, inflammmaging, is implicated in a wide range of age-related diseases [8]. Inflammation was considered part of the original hallmark “altered intercellular communication”, which encompasses the decline of the immune response (immunosenescence) accompanied by low-grade inflammation and possibly other intercellular interactions. In the “New” hallmarks of aging, inflammation is considered on its own merits because of its significant contribution to the aging process and crossplay with other hallmarks. Therefore, we can say aging is characterized by “inflammaging”.

Despite the preconception that cancer and aging in the context of cell senescence are antagonistic processes, studies have shown that changes in specific basic biological processes are shared in physiological aging and cancer [5,9,10]. The mechanisms in both cancer and aging underlie the time-dependent accumulation of cellular damage [5]. Aging is associated with several events at the molecular, cellular, and physiologic levels that influence carcinogenesis and subsequent cancer growth [9]. In turn, various cellular processes such as DNA damage responses and cellular senescence that serve as tumor suppressing mechanisms throughout life result in degenerative changes and contribute to the aging phenotype. Several hallmarks of aging [5] are shared with the hallmarks of cancer [10], e.g., genomic instability, epigenetic alterations, loss of proteostasis, mitochondrial dysfunction, cellular senescence, and altered intercellular communication. However, the shared mechanisms underpinning the two processes remain unclear. Therefore, it is crucial to understand the mechanisms underlying the hallmarks shared by aging and cancer, especially the sources of inflammaging in cancer [11,12].

## 3. Inflammaging: An Age-Related Driver of Cancer Progression

The immune system plays a vital role in recognizing and inhibiting malignant tissue growth. In the elderly population, individuals become more susceptible to opportunistic bacterial and viral infections and have an increased incidence of autoimmune diseases and malignancy [13]. There is an intricate interrelationship between inflammaging and immunosenescence, which are somehow identical and in other aspects very different, yet occurring in parallel, influencing each other mutually [14,15]. The organism’s immune history consists of a lifetime of constant activation of the innate immune response, producing pro-inflammatory mediators concomitantly with the direct antigenic challenges. These bouts of activation produce notable changes in the adaptive immune response. Inversely, the altered adaptive response contributes to the persistent stimulation of the inflammaging process by the lack of elimination of chronic activation, which leads to the development and progression of age-related chronic inflammatory diseases, including cancer. However, the exact role of this interplay between the innate and adaptive immune responses is not yet well defined, and it may not even be obvious in some cases.

Chronic inflammation plays a role in many diseases, in some of them more common and severe. For example, chronic inflammatory diseases contribute to more than half of deaths worldwide [16]. Inflammaging is a smoldering chronic pro-inflammatory phenotype in the elderly in the absence of viral infection due to over-activation and a decrease in the precision of the innate immune system. It is characterized by increased blood serum interleukin-6 (IL-6), IL-1, IL-8, and tumor necrosis factor-alpha (TNFα), and by a subsequent increase in primary inflammatory markers, such as C-reactive protein (CRP) and serum amyloid A (A-SAA) that accompany aging [17]. This low-level chronically-activated, generalized pro-inflammatory state interacts with the genetic background and environmental factors, potentially triggering the onset of the most significant age-related diseases, such as cardiovascular disease, atherosclerosis, metabolic syndrome, type 2 diabetes, obesity, neurodegeneration, arthritis, osteoporosis and osteoarthritis, sarcopenia, major depression, frailty and cancer [17,18]. Clinical trials have demonstrated that some nonsteroidal anti-inflammatory agents neutralize IL-1 and TNF and retard the manifestations of cardiovascular pathologies, supporting the role of inflammation in this age-related disease [19].

Inflammaging is often attributed to the aggressive activation of immune cells, which include innate immune cells, such as macrophages [20] and immune cells in the adaptive immune system. For example, T lymphocytes are a significant source of cytokines, which amplify or modulate the inflammatory response and alter the phenotypes of nearby cells, often to the detriment of normal tissue function. CD4^+^ T cells are an essential arm of the adaptive immune response. Upon activation, they differentiate into various subsets, including Th1 and Th2 cells, follicular helper (Tfh) cells, Th17 cells, and regulatory T cells (TREG). The CD4^+^ T cell compartment functions are diverse, ranging from the activation of both immune and nonimmune cells to direct cytolytic activity, and to dampening of the immune response. It has been widely accepted that aging is characterized by a pro-inflammatory imbalance of Th1/Th2 cells, which is observed in the significant age-related diseases [21,22,23,24,25]. Recent studies showed that aging is associated with an increased pro-inflammatory Th17 immune response and anti-inflammatory Treg numbers and functions. Therefore, the Th17/Treg balance is also disturbed during aging [26,27,28,29], and this may play a critical role in the aging process and in age-related diseases, including cancer.

## 4. Th17 Inflammaging Plays a Pivotal Role in Age-Related Cancer Progression, Lessons from Aging Animal Models

Th17 cells have been implicated in developing autoimmune and chronic inflammatory diseases in humans. Th17 cells are a pro-inflammatory subset, while Treg cells have an antagonistic effect. Their developmental pathways are reciprocally interconnected, and there is essential plasticity between Th17 and Treg cells. These features imply that the Th17/Treg balance plays a significant role in the development and disease outcomes of human autoimmune and inflammatory diseases. During these diseases, this balance is disturbed, and this promotes the maintenance of inflammation. There was a report that aged T cells manifest augmented IL-17 responses over time and in dose response compared to young T cells [30], and extended IL-17 responses recently described in the elderly are of considerable interest. De Angulo et al., using an aging mimic mouse model (GPAT-1 KO mice), found that aged T cells secrete higher IL-17 that activates pro-inflammatory NF-κB signaling in prostate cells in vitro [31,32]. Given the complexity of T cells, and considering that the impact of aging on T cell responsiveness is different among various T-cell subsets [33], it is crucial to define how T subset responses (CD4 and CD8 T cells) in the aging process promote or inhibit prostate cancer. In this regard, we have utilized C57BL/6J (B6) mice as a rodent aging model and studied the potentiation of Th17 cell response in prostate tumorigenesis during aging [34]. We found that the Th17 cell response and Th17/Treg ratio are elevated in aged mice. Naïve CD4^+^ T cells from aged mice differentiated into cells producing increased Th17. Upon anti-CD3/anti-CD28 stimulation, purified CD4^+^ T lymphocytes from aged (96–104 weeks-old) mice produce more Th17-associated cytokines (IL-17A, IL-17F, and IL-22) and transcription factors (RORγt and Foxp3) than cells from younger (16–20 weeks-old) mice. In addition, the Th17/Treg ratio (calculated using mRNA levels of RORγt and Foxp3) is significantly enhanced in aged mice compared to young mice. In a series of experiments performed on primary CD4^+^ T cells stimulated with anti-CD3/anti-CD28, we demonstrated that factors (conditioned media) secreted from aged mice CD4^+^ T cells promote prostate cancer cell viability, migration, and invasion [34].

Furthermore, we confirmed that aged mice have significantly increased prostate mass, inflammatory cell infiltration, and mRNA levels of Th17 cytokines and pro-inflammatory mediators (IL-6, TNFα, and IL1β). In addition, prostate tissues of aged mice have significantly activated NF-κB/ERK1/2 signaling compared to young mice. Therefore, Th17-related inflammation during aging (named Th17 inflammaging) increases in prostate tissues and plays a vital role in age-related morphologic alterations.

A recent report [35] on a human study uncovered a dominant Th17 inflammaging profile produced by CD4^+^ T cells. Furthermore, this study elucidated a plausible mechanism for the age-associated, metformin-treatment-associated Th17-linked cytokine hyper-producer phenotype. Specifically, CD4^+^ T lymphocytes from healthy older people preferentially produce a Th17 profile. The knockdown of autophagy in T cells from young subjects activates this profile. In vitro metformin improves autophagy and mitochondrial function in parallel to decreasing Th17 inflammaging. Therefore, CD4^+^ T cells from aged individuals exhibit defective mitochondrial autophagy, resulting in altered redox metabolism and the upregulation of Th17 cytokines, which may contribute to aging-associated chronic inflammation or inflammaging [35]. This result is consistent with our findings that Th17-related inflammation is at the center of inflammaging. We also found increased Th17-related cytokines in human prostate tissue in aged individuals. Therefore, it is crucial to understand how the Th17 inflammaging leads to the development of cancers associated with age. Our ongoing work is focused on the functional role of Th17 response and the imbalance of the Th17/Treg ratio [34] in prostate carcinogenesis during aging by using an autochthonous prostate cancer mouse model generated at different ages [36].

## 5. Novel Strategy for Generating Age-Related Cancer Models in Mice

Although there is a close relationship between aging and cancer, they are often investigated separately. The techniques for studying the impact of aging on cancer initiation and progression still need improvement. Mouse models and GEMM are essential experimental tools for understanding the molecular mechanisms of complex biological systems and related gene pathways. These tools may be significant in therapeutic strategies for studying aging or cancer. Recent reviews have summarized mouse models modeling aging and cancer [37,38]. These models generated to explore the signaling pathways manifest either aging or cancer phenotypes, sometimes both, which worked well for understanding the correlation between aging and cancer. However, few models reflect the natural aging process that contributes to the development of age-related cancers. Therefore, here, we focus on discussing a novel method that can generate age-related mouse cancer models at different ages, which could elucidate better how age impacts cancer development and progression [36].

We know that the Cre-loxP system is the most widely utilized, powerful technology for genetic manipulation in mammalian cells and experimental animals [39]. The Cre recombinase is a 38-kDa protein that recognizes and mediates site-specific recombination between loxP-site sequences in bacteriophage P1 [40,41]. The general principle of the Cre-loxP system is that the Cre recombinase recognizes two directly repeated loxP sites. The Cre excises the loxP flanked (floxed) DNA, thus creating two DNA products, a circular DNA with the excised gene, and an inactivated target gene [39]. Conditional mutagenesis using the Cre-loxP system in genetically modified mice is one of the essential tools to help us understand cellular or molecular interactions in in vivo biological models. This system has enabled researchers to investigate genes of interest in a specific tissue/cell (spatial control) and/or time (temporal control). The spatial regulation of recombination can be achieved by using cell-type-specific promoters that drive the expression of Cre in the cell population or tissue of interest. The temporal regulation can be obtained by either regulatable promoters or a small-molecule inducer that binds to a fusion protein consisting of Cre and the ligand-binding domain of steroid receptors. The CreER recombinase is one of the ligand-dependent Cre recombinases, consisting of Cre fused to the mutated hormone-binding domain of the estrogen receptor (ER). It is inactive but can be activated by the synthetic ER ligand tamoxifen or 4-hydroxytamoxifen. Improved versions of the chimeric Cre recombinase have been developed, including CreERT2 [42,43] and ERT2CreERT2 [44]. By combining the tissue-specific expression of a CreER recombinase with its tamoxifen-dependent activity, the Cre-mediated gene regulation can be spatially and temporally controlled. Various tissue-specific Cre-driver mouse lines have been generated, and new Cre lines continue to be developed [45,46].

To better define the impact of aging on prostate cancer development and progression, a spatially and temporally controlled *Pten* conditional knockout mouse model is necessary. Generating genetically engineered mice is a time-consuming knockout approach. Additionally, aging studies require an extended period of time to age the mice. Therefore, we developed a virus-assisted *Pten* conditional knockout model (named *Pten^adcre+^*) instead of generating a tamoxifen-inducible *Pten* knockout model [36]. Our model allows a comparison of tumor formation in the same time interval post-*Pten* deletion between aged and non-aged mice. The prostate-specific Cre-LoxP gene switching in this model is generated via intraductal delivery of an adenovirus expressing Cre-recombinase with luciferin tag (Adeno-Cre-Luc) to the prostate anterior lobes (please check Ref. [36] (Figure 1) and Figure 1 in this paper). The Ad-CMV-Luc is injected into wild-type mice as a control. This approach saves time and allows us to confirm the success of injection in the mouse prostate by live imaging after injection [36].

In this model, the *Pten^L/L^* young (12 weeks old), middle-aged (44 weeks old), and aged (96 weeks old) male mice were deeply anesthetized with 2–4% isoflurane in oxygen. All surgical procedures were performed under sterile conditions and per the guidelines of the Institutional Animal Care and Use Committee at Tulane University [36]. The mouse was shaved by a surgeon wearing a mask and sterile gloves, before a skin application of a betadine solution and 70% ethanol. An analgesic drug (Buprenorphine) was administered to the mouse before the incision. Then a tiny incision (1.5 cm) was made to expose the seminal vesicles (SVs). The SVs were exteriorized only briefly for injection, to prevent tissue dehydration. We conducted all injections within a biosafety hood with a modified sash since adenovirus was used. We used A LEICA S9D Stereo Zoom microscope to facilitate surgical maneuvers. We placed the anesthetized mouse on a heating pad during surgery, and 5–10 μL (4∼8 × 10^6^ PFU/g of body weight) of viral solution (1 × 10^10^ PFU/mL) was intraductally delivered into the anterior prostate (AP) using a micropipette (Hamilton Company, Reno, NV, USA). The Cre-expressing Adenovirus (Cat. No. 1705, named Ad-Luc-Cre) was obtained from Vector Biolabs, Malvern, PA. After injection, the peritoneum was sutured using a 5-0 vicryl suture, and the skin was closed using a 5-0 nylon suture (please check Figure 1 in [36] and Figure 1 in this paper). In our models, Cre recombinase is expressed explicitly in prostate epithelial cells but not stromal cells (see Figure 2 in [36]). This confirms that our novel method of generating the *Pten* knockout mouse model is successful and can be used to study adenocarcinoma developed from epithelial cells but not stromal cells. Further, the virus-assisted Cre-expression-mediated biallelic ablation of the *Pten* gene at different ages can initiate prostate carcinogenesis and closely mimics the course of human prostate cancer formation. We know that human prostate cancer development proceeds through several steps, including prostatic intraepithelial neoplasia (PIN), adenocarcinoma, and metastasis. In this model, the focal hyperplasia of epithelial cells occurs four weeks post-*Pten* ablation for all age groups. By eight weeks, the prostatic intraepithelial neoplasia (PIN) lesions have grown in the prostate epithelium, and some PINs progress to micro-invasive adenocarcinoma. By 16 weeks post-*Pten* ablation, the invasive adenocarcinoma occurs and increases (see Table 1 in [36]). The most important thing is that prostate carcinogenesis initiated in middle-aged mice and aged mice develops with significantly more rapid onset and progression than in young mice. By analyzing the cellular proliferation, apoptosis, and inflammatory cell infiltration (see Figures 5 and 6 in [36]), we demonstrated that the underlying mechanisms by which aged mice show increased incidence and progression of prostate cancer are related to increased cellular proliferation, decreased apoptosis, and increased inflammatory cell infiltration. All these phenomena mimic human prostate cancer development and progression [47].

## 6. Conclusions

In summary, the novel virus-assisted spatially and temporally controlled method to generate an age-related cancer model can address the role of aging in cancer initiation and progression (Figure 1). In addition, this model can be used to study specific genes in *Pten*-related carcinogenesis during aging by crossing these gene-specific knockout mice with the *Pten* floxed mice and then using a similar approach to precisely deliver Cre to specific tissues in a temporally manageable manner. Furthermore, in addition to *Pten*, other tumor suppressor genes can be knocked out at different ages to model the respective gene-related cancer. Investigators can borrow a similar approach to generate age-related models of solid cancers, just as we have for prostate cancer. In addition, these age-dependent cancer models can be used to test the efficacy of pharmacological agents in preventing and treating cancers in aged versus non-aged mice. Therefore, to better understand the effects of aging on tumorigenesis, the virus-assisted spatially and temporally controlled gene knockout mouse model will provide researchers a means to separate the effects of aging from the inherent properties of the tumor.

## Figures and Tables

**Figure 1 cancers-14-05185-f001:**
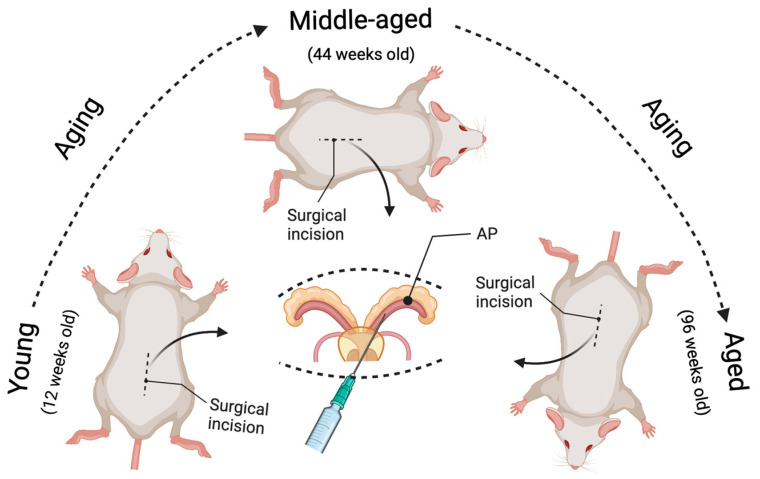
Illustration of the novel strategy to model age-related prostate cancer. The prostate-specific *Pten* knockout prostate cancer mouse model can be induced at different ages (young, middle-aged, and aged) by intraductal delivery of an adenovirus expressing Cre-recombinase to the prostate anterior lobes (AP) of *Pten* floxed mice. The surgical incision was performed at 12 weeks of age for the young group, 44 weeks for the middle-aged group, and 96 weeks for the aged group. As a result, the tumor growth can be compared in the same time interval post-*Pten* excision between the aged and non-aged mice (e.g., 4, 8, and 16 weeks post-delivery of the Cre-expressing adenovirus), leading to a better understanding of the effects of aging on prostate carcinogenesis.

## Data Availability

The data presented are available in the references cited in this article.

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
