# Peer review of "A Novel Strategy to Model Age-Related Cancer for Elucidation of the Role of Th17 Inflammaging in Cancer Progression"

_cancers, 2022, doi:10.3390/cancers14215185_

Round 1

Reviewer 1 Report

Title, abstract and key worlds may be better correspond with the manuscript - as the work is about animal model I would suspect the "animal model" worlds in the title.

The authors proposed mouse model strategy for prostate cancer elucudation of the role of Th17 inflammaging in the cancer progression.

The manuscript is narrative opinion and there are no methods strategy neither results described.

The references are quite old. only 15 of 45 refferences are from last 5 years. Moreover some references lack the authors or pages in the lines (eg.: line 303: Aging NIo: Understanding the Dynmics of the Aging Process. In.; 2021.)

I would sugesst to add methods strategy and to describe results.

Moreover please state the Bioetical Commitiee statement aproval. If the work did not need the aproval please state this.

Author Response

Response to Reviewer 1 Comments

Point 1: Title, abstract and key worlds may be better correspond with the manuscript - as the work is about animal model I would suspect the "animal model" words in the title.

Response 1: Thanks for your suggestion.  The terms “mouse model” are used explicitly in the Abstract and Key Words.  However, “Model” is used as a verb in the Title, so we refrain from adding “mouse” to the title.

Point 2: The authors proposed mouse model strategy for prostate cancer elucidation of the role of Th17 inflammaging in the cancer progression. The manuscript is narrative opinion and there are no methods strategy neither results described. 

Response 2: Thank you for mentioning this point. This article is submitted as a “Perspective or Opinion, and as such is reliant on literature references for much of the methods and results.  As we have published our novel animal model with a detailed description of methods and results, this paper focuses on the perspective for using this age-related cancer animal model. However, we have now added the surgical procedures and some results to let the audience clearly understand the strategy.

Point 3: The references are quite old. only 15 of 45 refferences are from last 5 years. Moreover some references lack the authors or pages in the lines (eg.: line 303: Aging NIo: Understanding the Dynmics of the Aging Process. In.; 2021.)

Response 3: Thank you for pointing out this. We have updated all the references and revised the vague references. Now, 30 out of the 47 references are from the last five years (2017 to 2022)

Point 4: I would suggest to add methods strategy and to describe results.

Response 4: Thank you for your suggestion. We have added the methods and described some important results.

Point 5: Moreover please state the Bioetical Commitiee statement aproval. If the work did not need the aproval please state this.

Response 5: Thank you for pointing this out. Yes, our animal model generation was approved by the IACUC, and we have the approval statement. We also added this information in the manuscript. Please check the text.

Reviewer 2 Report

Dr. Qiuyang Zhang and colleagues interestingly studied the novel strategy to model age-related prostate cancer. Further they explained the novel method to generate age-related cancer models in mice to better understand how age impacts cancer initiation and progression in the natural aging process.

 Minor Comments:

Figure 1: The authors should mention the number of days on surgical incision in detail.

Round 2

Reviewer 1 Report

The authors answered to the rewiever all points, however I still propose to change the title.

Other may be accepted

Reviewer 2 Report

I recommend the manuscript for acceptance and for further publication.